# Recombinant TP-84 Bacteriophage Glycosylase–Depolymerase Confers Activity against Thermostable *Geobacillus stearothermophilus* via Capsule Degradation

**DOI:** 10.3390/ijms25020722

**Published:** 2024-01-05

**Authors:** Beata Łubkowska, Ireneusz Sobolewski, Katarzyna Adamowicz, Agnieszka Zylicz-Stachula, Piotr M. Skowron

**Affiliations:** 1Faculty of Health and Life Sciences, Gdansk University of Physical Education and Sport, K. Gorskiego 1, 80-336 Gdansk, Poland; 2Faculty of Chemistry, Department of Molecular Biotechnology, University of Gdansk, Wita Stwosza 63, 80-308 Gdansk, Poland; ireneusz.sobolewski@ug.edu.pl (I.S.); kachapoll@gmail.com (K.A.); a.zylicz-stachula@ug.edu.pl (A.Z.-S.); piotr.skowron@ug.edu.pl (P.M.S.)

**Keywords:** *Geobacillus*, *Geobacillus stearothermophilus*, *Geobacillus thermoleovorans*, glycosyl hydrolase, depolymerase, bacteriophage, TP-84, thermophile, thermophage, capsule, envelope

## Abstract

The TP-84 bacteriophage, which infects *Geobacillus stearothermophilus* strain 10 (*G. stearothermophilus*), has a genome size of 47.7 kilobase pairs (kbps) and contains 81 predicted protein-coding ORFs. One of these, TP84_26 encodes a putative tail fiber protein possessing capsule depolymerase activity. In this study, we cloned the TP84_26 gene into a high-expression *Escherichia coli* (*E. coli*) system, modified its N-terminus with His-tag, expressed both the wild type gene and His-tagged variant, purified the recombinant depolymerase variants, and further evaluated their properties. We developed a direct enzymatic assay for the depolymerase activity toward *G. stearothermophilus* capsules. The recombinant TP84_26 protein variants effectively degraded the existing bacterial capsules and inhibited the formation of new ones. Our results provide insights into the novel TP84_26 depolymerase with specific activity against thermostable *G. stearothermophilus* and its role in the TP-84 life cycle. The identification and characterization of novel depolymerases, such as TP84_26, hold promise for innovative strategies to combat bacterial infections and improve various industrial processes.

## 1. Introduction

The TP-84 bacteriophage has a very narrow host range, infecting thermophilic *G. stearothermophilus* strain 10 and several other *Geobacillus* strains with much lower efficiency [1]. The *Geobacillus* genus consists of thermophilic bacteria that are capable of thriving within a temperature range of 35–75 °C, with an optimal growth temperature of 55–65 °C. Initially classified as members of the *Bacillus* genus, the *Geobacillus* species were later established as a distinct genus in 2001 [2]. These rod-shaped, Gram-positive, spore-forming bacteria respire aerobically or facultatively anaerobically [3]. Neutrophilic bacteria belonging to this genus can proliferate in a pH range of 6.0–8.5, with a most favorable pH of 6.2–7.5. Most *Geobacillus* species have relatively simple requirements and can grow without the need for growth factors or vitamins. Remarkably, they can utilize n-alkanes as both carbon and energy sources [4]. In addition, *G. stearothermophilus* is a thermophilic bacterium known for its ability to produce large-volume capsules composed of polysaccharides, which exceed the vegetative cell volume tens of times [3]. These capsules play a crucial role in protecting the bacterium from environmental stresses and host defense responses, including hindering the efficacy of antimicrobial agents and limiting the access of bacteriophages to their host cells. Thermophilic microorganisms can be easily obtained by incubating various environmental samples in conventional cultivation media at high temperatures [5]. This method was employed to isolate the prototrophic strain of *G. stearothermophilus* that could grow in a medium containing only glucose and mineral salts [3,4]. However, auxotrophic strains required additional supplements such as biotin, thiamine, nicotinic acid, and DL-methionine [6]. Furthermore, it was discovered that using a rich medium consisting of beef extract, soy peptone, and 0.2% NaCl buffered with K_2_HPO_4_/KH_2_PO_4_ resulted in 10-times-higher biomass production by *G. stearothermophilus* compared to the standard fermentation medium [7]. *Geobacillus* bacteria form spores, which has led to the gradual accumulation of high bacterial populations over long periods of time. The adaptive characteristics of *Geobacillus* spores allow them to disperse in the atmosphere and be transported over long distances. These spores remain viable for extended periods [8]. Spores of *Bacillus* species, which are closely related to geobacilli, have demonstrated resilience against heat, radiation, and chemicals [3,9]. The capacity to survive and thrive at high temperatures, as well as the ability to utilize and produce a wide range of compounds, make these bacteria and their byproducts highly desirable for applications in various fields, including the food, paper, biotechnology, medical, and bioremediation industries. Moreover, *Geobacillus* species serve as valuable sources of various thermostable enzymes, including proteases, amylases, and lipases, among others. They are capable of producing exopolysaccharides and bacteriocins, and they contribute to the production of biofuels and bioremediation processes. The wide range of applications for *Geobacillus* has sparked increasing interest in studying their physiological and biochemical characteristics, leading to the discovery of new areas where they can be utilized, such as in bioenergy production. The growing demand for energy has prompted the exploration of alternative technological options, and *Geobacillus* species have demonstrated their ability to generate or enhance the productivity of important bioenergy sources, including ethanol, isobutanol, 2,3-butanediol, biodiesel, and biogas. As a result, new applications are continuously emerging for this group of thermophilic bacteria [10].

Bacteriophages, especially those previously classified under the families *Podoviridae, Siphoviridae*, and *Myoviridae* [11], serve as significant sources of depolymerase enzymes. In 2021, the International Committee of Taxonomy of Bacterial Viruses (ICTBV; https://ictv.global/sc/bacterial (accessed on 31 December 2023).) proposed a novel classification [12,13], grouping bacteriophages from the aforementioned families under the class *Caudoviricetes* (https://ictv.global/taxonomy (accessed on 31 December 2023). Depolymerases (DPs) are typically found as structural proteins, such as tail fibers and baseplates, and can also exist as soluble proteins during the lytic cycle of the phage. They play a crucial role in the initial attachment of the phage to the host bacterium and are responsible for degrading the bacterial capsule, thereby initiating the process of phage infection [14,15]. Polysaccharide DPs have the ability to cause damage to bacterial cells, specifically targeting and degrading capsular polysaccharides, as well as structural polysaccharides, including exopolysaccharides, which are major components of bacterial biofilms. DPs can either be attached to a bacteriophage tail or exist in a free form, diffusing into the surrounding medium. DPs are classified into two groups: hydrolases (glycanases) or polysaccharide lyases [15,16]. These enzymes display a high degree of heterogeneity in terms of their substrate specificity, molecular weight, and sensitivity to physical and chemical factors. Bacteriophages that produce DPs have the ability to target encapsulated bacteria and hold great potential as a novel class of anti-biofilm agents. By degrading the bacterial capsule, DPs increase bacteria sensitivity to a bacteriophage infection. Furthermore, when applied in wound treatment, DPs reduce the virulence of the bacteria and sensitize them to the human immune system. Therefore, these enzymes show promise as potential therapeutic agents [17,18].

In this study, we have cloned, modified, and expressed the TP84_26 gene, coding for TP-84 DP, purified the recombinant protein, developed specific assays for its activity assessment, and characterized the enzyme. TP-84 is, thus far, the most characterized thermophilic bacteriophage infecting *G. stearothermophilus* [1,2,3,19]. The results shed some light on the mechanisms by which TP-84 targets cells and disrupts the capsules of *G. stearothermophilus*, providing potential insights for developing targeted therapeutic interventions against capsular bacteria.

## 2. Results and Discussion

### 2.1. Cloning of the TP84_26 DP and His6-TP84_26 Genes

We have successfully obtained expression clones of recombinant TP84_26 DP in two variants: (i) native ORF (Appendix A) and (ii) with an introduced modification at the N-terminus through the addition of His6-tag for Immobilized Metal Affinity Chromatography (IMAC) application (Appendix A), intended for the high-yield production of pure enzyme. The obtained enzyme variants may be helpful in bacterial biofilm removal from industrial facilities, among others. The His6-tag was fused with the TP84_26 ORF through cloning into our previous construct, pET21d-His, obtained through the modification of the pET21d(+) expression vector via site-specific mutagenesis, introducing segment coding for His6-tag and possessing a shifted NcoI restriction site, now following the His6-tag segment. Further, with the use of PCR mutagenic primers, the DP gene was cloned into the vector (Appendix A). The designed primers allowed for the incorporation of specific DNA sequences, recognized by the BsaI (Type IIS) and SalI restriction endonucleases, at the 5′ and 3′ gene ends, respectively. The use of the BsaI design enabled the directional cloning of the TP84_26 ORF in a seamless fusion with the pET21d(+) vector’s start codon in a way that involved no introduction of additional codons to maintain the recombinant enzyme native amino acid sequence, ensuring the preservation of its functional properties (Appendix A). As we showed previously [1], the codon usage in the TP-84 genome ORFs was very well suited for the *E. coli* expression, resulting in massive biosynthesis of the TP-84 recombinant proteins. Therefore, the codons of the TP84_26 gene did not require optimization, as they were already highly compatible with the *E. coli*/T7 expression system. Genetic maps and sequences of the expression constructs (illustrating the arrangement of genetic elements, nucleotide, and amino acid sequences) are shown in Appendix A. The cloning process yielded a substantial number of clones, and the cultures of these clones demonstrated continuous growth even after the induction of the recombinant gene expression, indicating that recombinant TP84_26 DP variants do not exert toxic effects on the *E. coli* host (Appendix A). The clone growth curves enabled the selection of the best clone suitable for the upscaling of protein biosynthesis, providing the highest enzyme yields upon optimization of the cultivation conditions.

### 2.2. Gene Expression/Purification of Recombinant His6-TP84_26 DP and TP84_26 DP and Determination of TP84_26 DP Localization within TP-84 Capsid

The native recombinant TP84-26 DP and the His6-tagged TP84_26 DP fusion variant were successfully expressed in *E. coli* and purified, with the native recombinant TP84_26 DP shown in Figure 1 and Figure 2. Interestingly, it was found that the recombinant His6-TP84_26 DP did not bind to the immobilized nickel ions on IMAC, despite the presence of the His6-tag. This suggests that the tag may be buried within the protein structure. Thus, a multi-step purification process suited to the purification of both His6-TP84_26 and native recombinant TP84_26 DPs was developed. The procedure included six steps: (i) the disruption of the recombinant *E. coli* using ultrasound, (ii) selective precipitation using PEI and ammonium sulfate from the crude protein extract, containing the expressed TP84_26 DP variant [3] (this step turned out to be very effective in removing the bulk of impurities such as nucleic acids and other proteins), (iii) Blue-Sepharose 6 FF affinity chromatography used in ‘negative’ mode, (iv) ion-exchange chromatography on SOURCE 15Q, a strong anion exchanger (Resource^TM^ Q column), (v) affinity chromatography on Heparin-Sepharose 6 FF, and (vi) ion-exchange chromatography on SOURCE 15Q. The TP84_26 depolymerase variants were purified to functional homogeneity. The presence of the protein band, corresponding to the native recombinant TP84_26, confirmed the successful purification (Figure 2a,b). The same procedure worked well on the His6-tagged recombinant TP84_26 DP. The resulting preparations were stored at 4 °C.

His6-tag is typically used as a fusion tag to facilitate purification through IMAC, as it has a strong affinity for immobilized metal ions such as nickel, manganese, or copper. However, in some cases, the tag may not be accessible due to its location within the protein three-dimensional structure [20]. Additionally, levels of immunodetection of the His-tagged protein can vary depending on the particular anti-His-tag antibody used [21]. In the case of His6-TP84_26 DP, one also needs to consider its high native molecular weight of 470 kDa and the apparent tetrameric structure (4 × 112 kDa polypeptide DP = 448 kDa, as we reported previously [3]), which may involve N-termini interactions. However, the tertiary composition estimate requires more in-depth confirmations, as known DPs are typically trimers. A crystal structure of the tail spike KP32gp38 DP, which degrades the capsule polysaccharide of *Klebsiella pneumoniae*, was resolved, and it was determined that it is a trimer with an extended, asymmetric shape [22]. This significantly increases the rotational radius of the KP32gp38 DP, which may also be the case for TP-84 DP, resulting in apparently much larger molecular size estimates during molecular sieving.

Moving the His6-tag to the C-terminus might help in IMAC purification, although this avenue was not further explored. The intact His6-TP84_26 DP reacted poorly on the Western blot with the anti-His6-tag antibodies (Figure 2d). This may indicate that the His6-tag epitope, recognized by the antibodies, is inaccessible in the intact polypeptide. The removal of SDS during Western blotting membrane washing in SDS-free buffer may partially result in the renaturation of the His6-TP84_26 DP. However, its large polypeptide size (112 kDa) may also contribute to incomplete protein transfer onto the Western blot membrane [23], thus decreasing the sensitivity of the Western blotting detection (Figure 2d).

Similar results were observed in the case of Western blotting with polyclonal anti-TP-84 antibodies (Figure 2a,b). The band corresponding to the native recombinant TP84_26 DP is very faint, while only a single proteolytic fragment (MW of app. 49 kDa), out of more than ten other fragments, gave a strong reaction (Figure 2b, lane 1; Appendix A). One can note that this band is prominent on the Western blotting membrane (Figure 2b, lane 1) despite only a trace amount of the fragment being detected in the duplicated Coomassie-blue-stained gel (Figure 2a, lane 1). This may suggest that only certain regions of TP84_26 DP associated with the TP-84 virion are exposed enough to induce antibody production, or that the enzyme itself has immunogenic epitopes hidden within the 3D structure. The association of TP84_26 DP with the bacteriophage TP-84 virion can be concluded based on the recombinant enzyme reaction with polyclonal antibodies developed against intact—not denatured or disrupted—highly purified TP-84 bacteriophages [3] (Figure 2b). Furthermore, this implies that the TP84_26 DP protein interacts with other viral components and is located in a specific region of the virion, allowing the accessibility of certain epitopes only. In our previous work [3], we showed that native TP84_26 DP is also produced in an unbound, soluble form, present in TP-84/*G. stearothermophilus* lysates. This would be biologically beneficial for the bacteriophage, allowing the soluble enzyme to soften the capsules from outside, while the virion-associated TP84_26 DP would ‘drill’ the capsule at the point of attachment to the host’s cells. These results also shed some light on the importance of conformation and structural context in immunoreactivity.

### 2.3. Development of an Enzymatic Assay for TP84_26 DP and Functional Implications

The TP84_26 DP ORF, bioinformatically detected in the TP-84 genome [1], was further confirmed using LC-MS and purification of the native enzyme [3]. A preliminary activity assay, based on Alcian blue dye reacting with polysaccharides, showed that native TP84_26 DP, isolated from the TP-84/*G. stearothermophilus* lysates, exhibits activity against *G. stearothermophilus* polysaccharides [3]. Here, we devised a more direct assay, evaluating the activity of the TP84_26 DP against TP84 capsules. To develop the assay, we used native recombinant TP84_26 DP (Figure 3) and His6-TP84_26 DP. In Figure 3, we present confirmation of the enzyme specificity through a biochemical assay designed to detect the degradation or stripping of the capsules from *G. stearothermophilus* cells. Streptomycin and sodium azide were added to the buffer during the cell substrate preparation to inhibit bacterial metabolism, prevent or limit the rebuilding of capsules, and stabilize the capsules’ digestion effect for the assay reliability (streptomycin blocks translation [24], while sodium azide blocks ATP synthesis in bacteria [25]). The investigated DP variant was added to the freshly prepared *G. stearothermophilus* cells’ substrate, and the samples were incubated for selected time intervals. Following the incubation, the samples were analyzed using Maneval’s negative staining and microscopic evaluation. Figure 3 shows the result of stripping the bacteria from the capsules, visualized as decreasing transparent capsule zones on a pink-violet background surrounding pink-violet-stained vegetative cells. The images captured during the assay time course provide direct visual evidence of the increasing effect of the enzymatic treatment of *G. stearothermophilus* capsules (Figure 3). The top control panels depict the images of thick *G. stearothermophilus* capsules without enzymatic treatment (Figure 3a–d), where no visible degradation or changes in the capsules’ shapes and sizes were observed. The internal panels show *G. stearothermophilus* capsules subjected to recombinant TP84_26 DP or His6-TP84_26 DP enzymatic treatment (Figure 3e–h). Both DP variants exhibited the same digestion pattern. After the addition of the DP, we observed the progressive degradation of the capsules over time. The enzymatic treatment leads to the breakdown, size decrease, and eventual removal of the capsules, resulting in visible changes to their structure and appearance. Furthermore, the heat-killed cells of the DP treatment shown in the bottom panels (Figure 3i–l) are more susceptible, with clearly visible vegetative cells completely stripped of capsules (green arrows). Figure 4 shows the disappearance of the quantitative capsules based on the analysis of Figure 3a–d.

The presented results strongly support the conclusion that the TP84_26 DP can strip bacterial capsules. However, considering the time needed for complete stripping (45 min), we conclude that either the reaction conditions need further optimization, or the residual capsule rebuilding activity of the pre-synthesized host’s enzymes confers concurrent activity in the case of using living cells. Interestingly, the substrate cells (and/or spores embedded within the cells) retained essentially the same rate of survival upon transient exposition to sodium azide and streptomycin combined with the removal of capsules (as determined by the bacterial titrations of *G. stearothermophilus* cells, taken from the samples shown in Figure 3, panels d–h) (Appendix A). Regardless of the streptomycin/azide effect, the survival of *stearothermophilus* upon capsule digestion indicates that they are not critical for the bacteria’s survival under ‘standard’ growth conditions, but may be crucial if the bacteria encounter an invading bacteriophage. These findings regarding ‘optional’ capsules corroborate our previously published results showing that the TP-84 ‘halo’ production is temperature-dependent [3]. While the developed assay confirms the TP84_26 DP/His6-TP84_26 DP capsule-stripping activity, it does not provide information about the enzyme’s chemical specificity. We attempted to acquire some insight into the chemical nature of the polysaccharide bonds cleaved by the TP84_26 DP using various commercial substrates: 4-Nitrophenyl β-D-glucopyranoside, 4-Nitrophenyl α-D-glucopyranoside, and 4-Nitrophenyl β-D-glucuronide. None worked, which indicates that TP84_26 DP is very specific toward *G. stearothermophilus* capsules. Overall, the presented results of the functional assay and biochemical/genetic analyses contribute to our understanding of the biological role of TP84_26 DP and its potential applications in capsule-, biofilm-, and thermophile-related research and biotechnology [26,27].

## 3. Materials and Methods

### 3.1. Bacterial Strains, Bacteriophage, Reagents

The TP-84 bacteriophage and *G. stearothermophilus* strain 10 were obtained from Piotr Skowron’s collection (the phage and its host were originally obtained from Epstein and Campbell [19]). The host is also available from the Bacillus Genetic Stock Center (BGSCID 9A5) (Columbus, OH, USA), although the BGSC strain exhibits minor differences in the genome. The streptomycin-resistant mutant of *G. stearothermophilus* 10 (*G. stearothermophilus* 10 strR; constructed by us previously [3]) was used in this work for biotechnology applications to minimize the possibility of scaled-up culture contaminations. For the TP84_26 cloning and expression, *E. coli* DH5a and *E. coli* BL21-Gold(DE3) were used, respectively. TP-84 bacteriophage propagation was conducted in TYM supplemented with streptomycin in liquid cultures or on agar plates [3]. The determination of titers was performed using an agar overlay procedure involving single plaque isolation, elution, and re-plating techniques. The protein standard PageRuler™ Plus Prestained Protein Ladder (cat. no 26619) and Coomassie™ Brilliant Blue were obtained from Thermo Fisher Scientific (Waltham, MA USA). A Resource^TM^ Q column for high resolution ion exchange chromatography and chromatographic resins, such as Ni Sepharose^TM^ 6 Fast Flow and Heparin-Sepharose™ 6 Fast Flow, were obtained from GE Healthcare Bio-Sciences AB (Uppsala, Sweden). The Vivaspin Turbo 15 PES (cat. no VS15T02) and VivaSpin^®^ Turbo 15 RC (cat. no VS15T41) centrifugal devices were from Sartorius Stedim Biotech GmbH (Goettingen, Germany). SIGMAFAST™ Protease Inhibitor Cocktail Tablets, EDTA-Free, and other chemicals were obtained from Merck KGaA (Darmstadt, Germany).

### 3.2. Cloning of TP84_26 DP ORF into E. coli

The TP84_26-coding gene was PCR-amplified from the TP84 genome upon the reaction optimization using various annealing temperatures (Appendix A). Mutagenic primers introduced BsaI and SalI recognition sequences flanking the 5′ and 3′ ends of the TP84_26 ORF (Appendix A). After BsaI and SalI cleavage, the PCR product was purified and ligated into the pET21d(+) vector or the modified pET21d(+)_His6 vector. Both vectors were previously cut with NcoI and SalI and gel-purified. The modified pET21d(+)_His6 vector already contained a His6-tag-encoding DNA sequence following the START codon (Appendix A). Two protein variants were obtained: (i) the native recombinant TP84_26 DP (Appendix A) and the His6-TP84_26 DP (Appendix A) with a His6-tag at its N-terminus for subsequent purification and identification using IMAC and Western blotting.

### 3.3. TP-84 Genetic Expression of His6-TP84_26 and TP84_26 Clones

Here, 500 µL of the selected *E. coli* [pET21d(+)_His6-TP84_26] or *E. coli* [pET21d(+)_TP84_26] overnight cultures, grown in LB media (1% tryptose, 0.5% yeast extract, 1% NaCl), was inoculated into six 250 mL flasks, each containing 50 mL of TB medium (tryptone 12 g/L, yeast extract 24 g/L, glycerol 4 mL/L, 0.017 M KH_2_PO_4_, and 0.072 M K_2_HPO_4_). Carbenicillin was added to the medium to a final concentration of 100 µg/mL, and the cultures were grown with vigorous aeration for 2.5–3 h until the OD600 nm reached 0.5, maintaining the logarithmic growth phase. The recombinant gene expression was induced with 1 mM isopropyl β-D-1-thiogalactopyranoside (IPTG), and the bacterial cultures were allowed to proceed for another 2.5–3 h (Appendix A). The bacterial biomass was centrifuged at 4 °C and the resulting pellets were frozen at −20 °C. Additionally, 1 mL samples were collected before the addition of IPTG and before the completion of the bacterial culture. The samples were centrifuged for 5 min at 4 °C and the pellets were frozen at −20 °C for further analysis.

### 3.4. Recombinant TP84_26 DP Variant Purification

#### 3.4.1. His6-TP84_26 IMAC Affinity Purification

The enzyme was subjected to purification via IMAC, using immobilized Ni^2+^ ions. Crude cell extract was prepared by resuspending the cells in 5 mL of the Q20 buffer (50 mM Tris-HCl pH 8.0 at 4 °C, 20 mM NaCl, 5% glycerol, 5 mM ß-mercaptoethanol (ßMe), 0.01 Triton X-100, 1 mM PMSF), the addition of 250 µL of the lysozyme stock solution (10 mg/mL), incubation for 1 h at 4 °C, and sonication. The suspension was centrifuged at 14,000× *g* for 10 min at 4 °C, and the supernatant was loaded onto a 1 mL IMAC Ni-Sepharose 6FF column containing immobilized Ni^2+^ ions. The column was equilibrated in the buffer NI (50 mM Tris-HCl pH 8.0 at 4 °C, 500 mM NaCl, 5% glycerol, 3 mM ß-Me, 0.01% Triton X-100). The column was washed with 3 mL of NI buffer and the protein was eluted using 2 mL (50, 100, 200, 350, and 500 mM) imidazole steps in the NI buffer. Two 1 mL fractions were collected for each imidazole concentration and analyzed using SDS-PAGE for the presence of recombinant His6-TP84_26 DP.

#### 3.4.2. Universal TP84_26 DP and His6-TP84_26 DP Purification Protocol

A multi-step purification process, omitting the His6-tag/IMAC application, was developed as scaled-up purification. Recombinant bacteria cultivation and initial processing were performed as described in the IMAC protocol version, except that two 1 L cultures in 5 L flasks were carried out until the OD600 nm reached 1. The bacterial biomass (10.6 g) was resuspended in the Q20 buffer (30 mM Tris-HCl pH 8.0 at 4 °C; 20 mM NaCl, 0.5 mM EDTA, 0.01% Triton, 0.01% Tween, 5% glycerol, 5 mM βMe, 0.5 tablet of the SigmaFast^TM^ protease inhibitor), and lysozyme was added to 0.5 mg/mL. The suspension was incubated for 20 min on ice, sonicated, and centrifuged for 30 min at 15,000× *g* at 4 °C. Then, 10% polyethyleneimine (PEI) (pH 8.0) was added to the supernatant to the final concentration of 0.5%. The mixture was stirred at 4 °C for 1 h and centrifuged for 20 min at 10,000× *g*. The pellet was resuspended in 50 mL of the Q250 buffer (with 250 mM NaCl), stirred at 4 °C for 45 min, and centrifuged for 20 min at 10,000× *g* at 4 °C. Proteins were precipitated from the supernatant with ammonium sulfate (0.5 g/mL), stirred for 45 min at 4 °C, and centrifuged for 20 min at 15,000× *g* at 4 °C. The precipitate was suspended in 25 mL of the Q20 buffer and dialyzed against Q20 buffer. The solution was centrifuged to remove the residues of insoluble proteins, and the supernatant (30 mL) was loaded onto 7.5 mL Blue Sepharose, used in ‘negative’ mode. The flow-through was loaded onto a Resource^TM^ Q column for high-resolution ion exchange chromatography and developed using a gradient composed of A20 buffer (30 mM Tris-HCl pH 8.0 at 4 °C, 0.5 mM EDTA, 20 mM NaCl, 0.01% Triton X-100, 0.01% Tween 20, 5 mM βMe) and B1000 buffer (30 mM Tris-HCl pH 8.0 at 4 °C, 0.5 mM EDTA, 1000 mM NaCl, 0.01% Triton X-100, 0.01% Tween 20, 5 mM βMe). The peak of the DP enzyme elution (obtained at 265 mM NaCl) was collected, dialyzed against the Q20 buffer, and loaded onto a Heparin Sepharose column (5 mL) equilibrated with the Q20 buffer. The column was washed with 20 mL of Q0 buffer (no NaCl added), and the enzyme was eluted with 10 mL of Q30 buffer (with 30 mM NaCl) and Q50 buffer (with 50 mM NaCl). Most of the enzyme eluted at 30 mM NaCl. Then, NaCl was added to the protein preparation (7 mL) to a final concentration of 60 mM, and the sample was loaded again onto a Resource^TM^ Q column. The proteins were eluted using a gradient composed of A60 buffer (30 mM Tris-HCl pH 8.0 at 4 °C, 0.5 mM EDTA, 60 mM NaCl, 0.01% Triton X-100, 0.01% Tween 20, 5 mM βMe) and B1000 buffer.

### 3.5. Immunoblotting

Western blotting was performed to detect the expressed TP84_26 or His6-TP84_26 DP variants using our custom-made rabbit primary anti-TP-84 antibodies [3] or primary rabbit monoclonal anti-His-tag antibodies (cat. no SAB5600227-100G, Merck), and secondary donkey polyclonal anti-rabbit s antibodies, labeled with peroxidase (cat. no SA1-200, ThermoFisher Scientific). Semi-dry transfer was conducted onto a PVDF membrane, pre-washed in methanol, deionized water, and then with Towbin transfer buffer (25 mM Tris-HCl pH 8.3, 192 mM glycine, 20% methanol, 0.04% SDS). The transfer was performed in the Trans-Blot Turbo^TM^ transfer system (Biorad Laboratories, Berkeley, California, USA) for 25 min at 25 V and 2.5 A. After transfer, the membranes were washed three times with TBS-T buffer (50 mM Tris-HCl pH 7.5; 150 M NaCl, 0.05%), incubated with EveryBlot Blocking Buffer (Bio-Rad) for 10 min at room temperature, and washed three times with TBS-T. Then, the membranes were incubated with either a solution of anti-TP-84 rabbit primary antibodies (1:500, 40 μL of antibodies added to 20 mL of TBS-T) or a solution of anti-His-tag antibodies for 1 h at 37 °C with gentle agitation. Following three washes with TBS-T, the membranes were placed in a solution of donkey anti-rabbit secondary antibodies (1:1000, 20 μL of antibodies added to 20 mL of TBS-T) for 1 h at 37 °C with gentle agitation (50 rpm). Finally, the membranes were washed three times with TBS-T, and a development solution (15 mL of TBS-T containing 10 mg of 3,3′-diaminobenzidine tetrahydrochloride (DAB) and 12 μL of 30% H_2_O_2_) was added.

### 3.6. Functional Enzymatic Assay

A novel DP functional assay was developed based on in vitro capsules stripped from intact *G. stearothermophilus* bacterial cells. The assay was dedicated to thermo-stable His6-TP84_26 DP and TP84_26 DP, at the optimal growth temperature of 55 °C for the bacterial substrate cells. The substrate for the enzyme was prepared using streptomycin-sensitive, native *G. stearothermophilus* 10 cells (100 mL of the culture) that were grown to the early log phase (OD_600nm_ = 0.3) and then centrifuged (5000× *g*, 10 min, 10 °C), subjected to two washes with cold TMC buffer (50 mM Tris–HCl pH 7.5 at 20 °C; 10 mM MgCl_2_, 5 mM CaCl_2_), supplemented with streptomycin (100 mg/mL) and sodium azide (1 mM). The cells were resuspended in 5 mL of the same buffer, aliquoted, frozen in liquid N_2_, and stored at −80 °C. Digestion reaction variant 1: The bacteria were treated with the recombinant His6-TP84_26 DP or recombinant TP84_26 DP (4 μg, 10 μL preparation added at 4 mg/mL in TMC buffer) at 55 °C in 100 μL of the cell substrate for time intervals of 0, 5, 15, and 45 min with shaking at 300 rpm to prevent cell sedimentation. Digestion reaction variant 2: As in variant 1, with additional heat treatment at 90 °C for 20 min prior to the DP digestion. Following the treatment, the capsules were detected using Maneval’s negative staining method and evaluated using light microscopy to assess the degree of capsule stripping. The determination of the capsules’ sizes was conducted upon observation (1600× magnification) through a microscopic ocular scale, photographing the microscopic image, enlarging the image, and measurement. This allowed for a quantitative evaluation of the capsules’ digestion progress (Figure 4).

## 4. Conclusions

The recombinant TP84_26 DP and His6-TP84_26 DP variants were purified using a ‘classic’ chromatographic approach due to the apparent inaccessibility of the His6-tag.

The Western blotting analysis revealed that TP84_26 DP is structurally associated with TP-84 bacteriophage particles.

A functional enzymatic assay was developed based on the TP84_26 DP treatment of a prepared substrate from intact *G. stearothermophilus* cells, employing Maneval’s negative staining and microscopic evaluation.

TP84_26 DP seems a viable thermostable candidate for further development in applications requiring the removal of bacterial biofilms in various fields, including biotechnology, medicine, or the food industry.

## Figures and Tables

**Figure 1 ijms-25-00722-f001:**
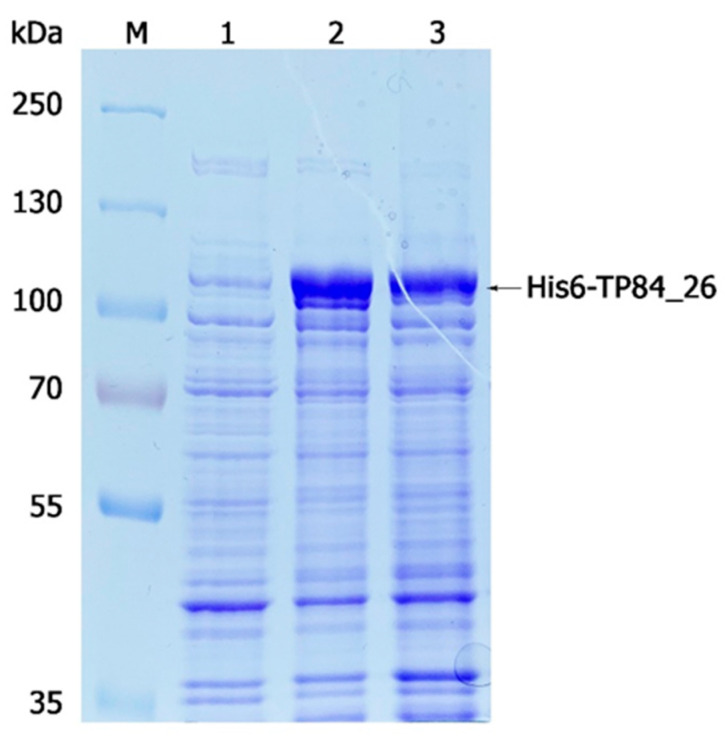
Optimization of expression of the recombinant His6-TP84_26 protein-coding gene analyzed with 8% SDS-PAGE using the selected recombinant *E. coli* [pET21d_His6-TP84_26] clone. Lane M, PageRuler™ Plus Prestained Protein Ladder; lane 1, uninduced culture (control); lane 2, 2.5 h after induction of the recombinant gene expression with IPTG; lane 3, 3 h after induction.

**Figure 2 ijms-25-00722-f002:**
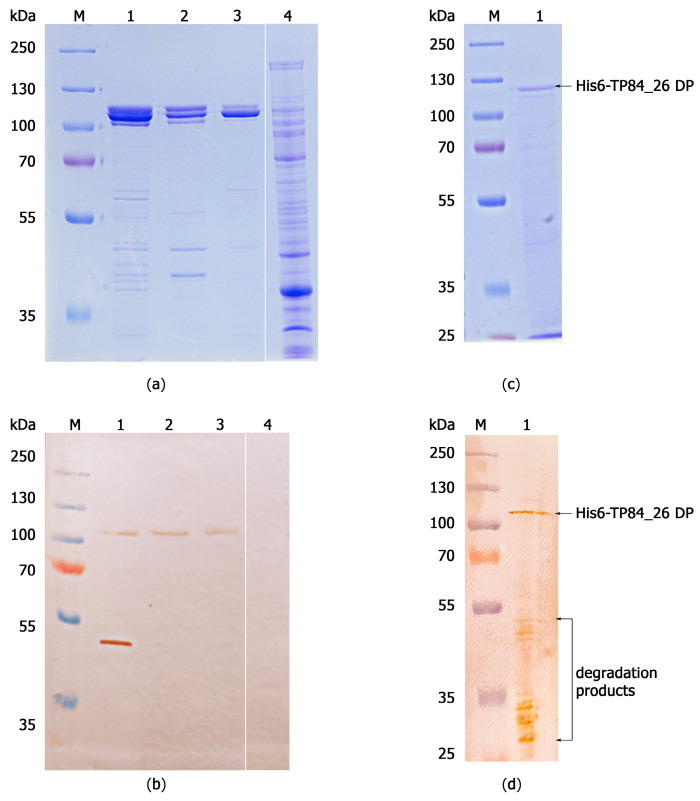
SDS-PAGE analysis and immunodetection of the recombinant TP84_26 DP variants. (**a**) SDS-PAGE analysis of the native recombinant TP84_26 DP in 10% polyacrylamide gel. Lane M, PageRuler™ Plus Prestained Protein Ladder; lane 1, final preparation of the native recombinant TP84_26 DP after the second Resource^TM^ Q purification step; lane 2, side chromatographic fractions (also containing the TP84_26 DP) from the second Resource^TM^ Q purification step (missing the 49 kDa enzyme fragment detected in panel (**b**), lane 1); lane 3, *E. coli* BL21-Gold(DE3) lysate (control for anti-TP-84 antibodies cross-reactions); (**b**) Western blotting and immunodetection of the native recombinant TP84_26 protein, using rabbit antibodies against TP-84 bacteriophage particles. The samples loaded and electrophoresis conditions are the same as in panel (**a**); (**c**) detection of the recombinant His6-TP84_26 DP in bacterial pellet by SDS-PAGE and Coomassie staining; (**d**) Western blotting and immunodetection of the His6-TP84_26 protein in recombinant bacteria pellet using anti-His6-tag antibodies. Lane M, PageRuler™ Plus Prestained Protein Ladder; lane 1, *E. coli* [pET_His6-TP84_26] cells, collected 3 h after induction of the recombinant gene expression with IPTG.

**Figure 3 ijms-25-00722-f003:**
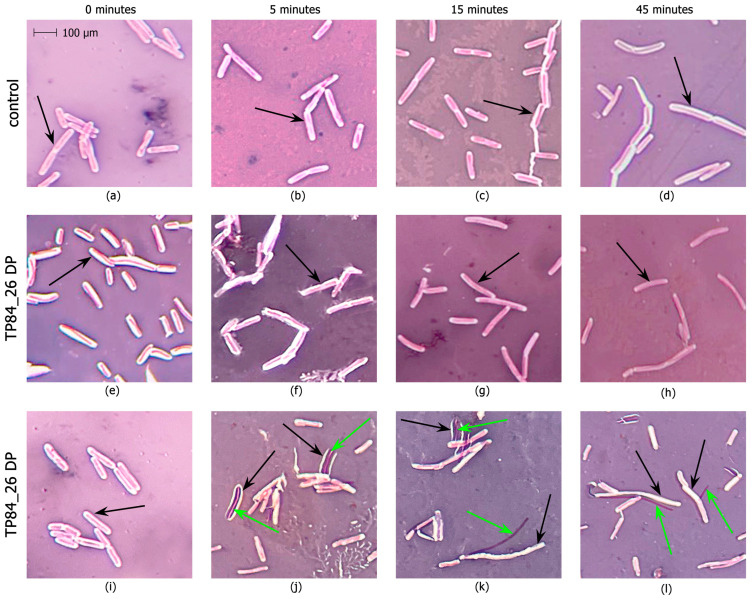
Capsules stripped from *G. stearothermophilus* cells using the developed Maneval functional assay. Microscopic evaluation of stripping of the bacterial capsules: control panels (**a**–**d**), thick capsules of *G. stearothermophilus* without the enzymatic treatment, observed at 0, 5, 15, and 45 min; panels (**e**–**h**), the effect of the enzymatic treatment of *G. stearothermophilus* cells with TP84_26 DP, observed at 0, 5, 15, and 45 min; panels (**i**–**l**), the effect of the enzymatic treatment of heat-killed *G. stearothermophilus* cells with TP84_26 DP, observed at 0, 5, 15, and 45 min. Progression of the capsule digestion and vegetative cell release is marked with black arrows (capsules) or green arrows (vegetative cells).

**Figure 4 ijms-25-00722-f004:**
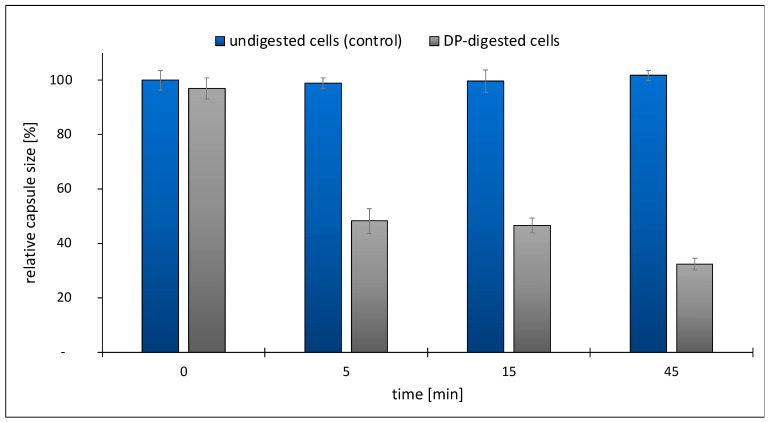
Quantitative evaluation of **c**apsules stripped from *G. stearothermophilus* cells using the developed Maneval functional assay, based on the results shown in Figure 3a–d.

## Data Availability

The authors confirm that the data supporting the findings of this study are available within the article and its Appendix A.

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
