# Peer review of "Recombinant TP-84 Bacteriophage Glycosylase–Depolymerase Confers Activity against Thermostable Geobacillus stearothermophilus via Capsule Degradation"

_ijms, 2024, doi:10.3390/ijms25020722_

Round 1
Reviewer 1 Report
Comments and Suggestions for Authors
The manuscript of Lubkowska et al. presents the synthesis and characterization of recombinant polysaccharide depolymerase from Geobacillus phage T84. The Authors well describe the cloning/expression/purification of the protein, noting difficulties with the application of N-terminal His-tagging. The existence of the protein as a part of a phage particle and the shielding of the tag by were shown by immunoblotting. Capsule depolymerization activity of the protein was monitored by Maneval staining with optical control. Generally, the narration is consistent and logical, well illustrated, and the paper can be considered for publication. Some notes:
Line 49 «host immune response» and line 96 «sensitized for immune system» — Geobaccilus is non-pathogenic environmental microorganism, and, as far as I know, no macroorganism is known to serve as a host for it;
Line 79 — modern phage taxonomy does not regard Myoviridae, Podoviridae and Siphovididae as families. Please change to Caudoviricetes if you wish to discriminate from Tectiviruses and filamentous phages.
Line 90 — Please present a reference concerning the induction of phage depolymerase in presense of polysaccharides. This statement seem arguable;
Line 151 — what is the sense of heparin sepharose step in the purification protocol?
T84_26DP is stated as tetrameric. The Authors have shown that by gel-permeation chromatography in their previous paper (Lubkowska 2023, Microbial Cell Factories). However, almost all studied phage polysaccharide depolymerases derived from phage adsorption apparatus are trimers. Maybe, it is worth to reconsider the statement?
Line 260 - typo, biofilms
Line 347 — Concentrations of ammonium sulfate in protein precipitation protocols are usually presented in % w/v;
Lines 396-400 — Since the Authors state the development of the method to monitor the capsule dissociation activity and state it to be semi-quantitative, it is advised to present more details of the protocol, eg. conditions of protein treatment: DP concentration, temperature, microscopy magnification, recording and processing the data on capsule size.
Lines 452-458 — References 3 and 4 seem to be the same. Please, format the reference list according to MDPI standards
Author Response
Dear Reviewer 1,
Thank you for revision my work. Your Comments and Suggestions helped me to improved my manuscript.
The requested reduction of self-citations and increasing other citations number have been implemented.
Also, please note, that the title has been slightly modified, to avoid repeating the work ‘capsule’.
Thank you.

Reviewer 2 Report
Comments and Suggestions for Authors
In this study, Łubkowska et al. focused on characterizing the TP84_26 DP, a putative tail fiber protein with capsule depolymerase activity in TP-84 bacteriophage. The researchers developed a direct enzymatic assay to test depolymerase activity and showed that the recombinant TP84_26 DP protein could break down the capsules of G. stearothermophilus. This study provides valuable insights into the function and mechanisms of bacteriophage's glycosylase-depolymerase. However, several issues need to be addressed before publication.
Major issue:
Figure 3: Please provide more solid data to support the conclusion. For instance, using a depolymerase dead mutant of TP84_26 DP as a control, or testing the concentration-dependent effect of the recombinant TP84_26 DP on degrading G. stearothermophilus capsules. Additionally, please use the arrow to indicate the difference between capsules after TP84_26 DP treatment, it also would be better if the authors could provide a quantitative evaluation for Figure 3.
Minor issues:
Line 90-91: “DPs are classified into two groups: hydrolases (glycanases) or polysaccharide lyases. ” Please provide a reference here.
Line 110 and 394: “IMAC” and “TMC”, when referring to something for the first time, please provide their full name.
Line 204: Please verify the lane labels for lanes 1 to 4 as they seem to be labeled incorrectly.
Author Response
Dear Reviewer 2,
Thank you for revision my work. Your Comments and Suggestions helped me to improved my manuscript.
The requested reduction of self-citations and increasing other citations number have been implemented.
Also, please note, that the title has been slightly modified, to avoid repeating the work ‘capsule’.
Thank you.

Round 2
Reviewer 2 Report
Comments and Suggestions for Authors
The authors have addressed my concerns, and the revised manuscript has been improved. I support the publication of this manuscript.